# Unique Nucleotide Polymorphism of African Swine Fever Virus Circulating in East Asia and Central Russia

**DOI:** 10.3390/v16121907

**Published:** 2024-12-11

**Authors:** Roman Chernyshev, Ali Mazloum, Nikolay Zinyakov, Ivan Kolbin, Andrey Shotin, Fedor I. Korennoy, Alexander V. Sprygin, Ilya A. Chvala, Alexey Igolkin

**Affiliations:** 1Federal Centre for Animal Health, 600901 Vladimir, Russia; zinyakov@arriah.ru (N.Z.); kolbin@arriah.ru (I.K.); shotin@arriah.ru (A.S.); korennoy@arriah.ru (F.I.K.); sprygin@arriah.ru (A.V.S.); chvala@arriah.ru (I.A.C.); igolkin_as@arriah.ru (A.I.); 2Department of Pathobiological Sciences, Louisiana State University, Baton Rouge, LA 70803, USA; ali.mazloum6@gmail.com

**Keywords:** molecular epidemiology, African swine fever virus, Asia, genotype II, single nucleotide polymorphisms

## Abstract

The lack of data on the whole-genome analysis of genotype II African swine fever virus (ASFV) isolates significantly hinders our understanding of its molecular evolution, and as a result, the range of single nucleotide polymorphisms (SNPs) necessary to describe a more accurate and complete scheme of its circulation. In this regard, this study aimed to identify unique SNPs, conduct phylogenetic analysis, and determine the level of homology of isolates obtained in the period from 2019 to 2022 in the central and eastern regions of Russia. Twenty-one whole-genome sequences of genotype II ASFV isolates were assembled, analyzed, and submitted to GenBank. The isolates in eastern Russia form two clades, “Amur 2022” and “Asia”. Within the latter clade, five subclusters can be distinguished, each characterized by a unique set of SNPs and indels. The isolates from the central regions of Russia (2019; 2021) form the “Center of Russia” clade, with two subclusters, “Bryansk 2021” and “Center of Russia 2021” (bootstrap confidence index = 99). The presence of the previously unique genetic variant ASFV for the Kaliningrad region in the wild boar population of the Khabarovsk region (eastern Russia; 2021) has also been confirmed.

## 1. Introduction

African swine fever (ASF), caused by the African swine fever virus (ASFV), is an emerging transboundary disease that infects domestic pigs and wild boars with a fatality rate of about 100%, causing enormous damage to the pig industry and hunting trade [1,2,3]. For example, in Russia and Eastern European countries, 800,000 pigs were destroyed between 2014 and 2017 [4]. In China, the total economic losses from ASF in 2019 amounted to 0.78% of the national gross domestic product [3]. The Georgian ASF pandemic of 2007 offered new alternative approaches to global and national ASF surveillance and control programs [5,6]. One of these approaches is based on molecular epidemiology [7], mainly aimed at identifying phylogenetic relatedness between isolates and detecting distinct ASFV genetic variants circulating within a certain territory [8]. Isolate differentiation and the spatiotemporal pattern understanding of circulating ASFV are currently indispensable parts of epidemiological surveillance in determining the territorial origin and transmission mechanisms, predicting the situation, and improving disease control measures [8].

Whole genome analysis demonstrates the highest resolution for such purposes; however, the “multi-gene” approach also allowed the identification of at least 24 clusters among genotype II ASFV isolates registered in Eurasia [9,10]. The C-terminal fragment of the *B646L* gene, encoding major capsid protein p72, is a traditional target for ASFV genotyping [11]. This region classifies all the studied isolates into 24 genotypes, including those recently recovered in Ethiopia, Mozambique, and Malawi [12,13]. The vast majority of ASF cases in Europe and Asia are caused by genotype II, characterized by genetic and biological conservation [14,15]. However, the ASF outbreaks reported on the Island of Sardinia (Italy) from 1978 to 2019 were associated with genotype I [16]. Moreover, in 2021, ASFV genotype I was isolated in the Henan and Shandong Provinces of China [17]. In 2023, for the first time in history, a recombinant ASFV variant (isolated in 2021) of genotypes I and II was detected in three provinces of China (Jiangsu, Henan, and Inner Mongolia). The same recombinant was later confirmed in the Russian East and the northern regions of Vietnam [18,19,20].

In Eurasia, unlike African countries, ASFV does not undergo a sylvatic transmission cycle (a cycle involves soft ticks (*Ornithodoros*), warthogs, and domestic pigs). Therefore, the ASFV genome is more conserved, making it difficult to differentiate isolates and trace back outbreaks [21]. Nevertheless, single nucleotide polymorphisms (SNPs) have been characterized at several loci (central variable region (CVR) of *B602L* gene, IGR *I73R/I329L*, *K145R*, *MGF 505-9R/10R*, and others) [22,23,24]. Such fragments are defined as “genome markers of transmission” and subsequently used for sub-genotyping [25,26]. The CVR and intergenic region *I73R/I329L* remain standard markers for ASFV differentiation [27,28]. However, the pattern of circulation of IGR variants in Russia has been inconclusive since 2018 [29]. This highlights the need to search for, analyze, and include other genome fragments in clustering isolates. For this purpose, higher-resolution assays (for example, high-throughput sequencing) are needed to discriminate sequences.

In late summer 2018, ASFV genotype II was introduced to China; in January 2019, it was introduced to Mongolia [30,31], and despite the enormous number of outbreaks, there are very few publicly available data on whole genome analysis of isolates from these countries [17,18,32,33,34,35]. Analysis based on data from Genbank demonstrated that the isolates from the Amur and Primorsky regions of Russia, which share a long border with China, are closely related to East Asian variants of the ASFV, as well as with the isolates from Belgium, Hungary, Moldova, and the Czech Republic [8,36]. The whole genome analysis of the virus, which has been endemic to the Russian East since 2021, has not been carried out. Therefore, it is difficult to assess the scale of circulation of different genetic variants, especially recombinant ones, which are rapidly spreading across Asian countries (China, Russia, and Vietnam).

The lack of data on whole genome sequences is a significant barrier to our understanding of genotype II ASFVs’ molecular evolution. As a result, it limits the range of detection of SNPs required for a more accurate and complete description of the pathogen’s circulation. The purpose of this study was to detect SNPs, perform a phylogenetic analysis, and analyze the level of homology among isolates recovered between 2019 and 2022 in the central and eastern regions of Russia.

## 2. Materials and Methods

### 2.1. ASFV Isolates

As part of the Federal Monitoring Program for ASF in Russia and to confirm laboratory diagnosis of ASF, spleen samples from dead wild boars and domestic pigs were submitted to the FGBI “ARRIAH” reference laboratory for ASF. A 10% suspension (*w*/*v*) was prepared from samples using a phosphate-buffered saline (PBS) solution; suspensions were then clarified by centrifugation at 1000× *g* for 3 min, passed through a 0.45 micropore size bacterial filter (Corning, Corning, NY, USA), and the resulting filtrate was used to infect porcine bone marrow primary (PBMP) cell culture. The ASFV was isolated in PBMP cell culture according to the previously published protocol [37]. All ASFV isolates showed hemadsorption, and the virus titer in PBMP cell culture was higher than 6.0 lg HAD_50_/cm^3^. Detailed epidemiological information (name of the isolate, animal of origin, sampling date, place, and titer) of 21 samples included in this study is provided in Appendix A, and the geographic distribution of those isolates is demonstrated in Figure 1.

### 2.2. Virus Genomic DNA Extraction

For each isolate, 40 mL of PBMPs were infected at a low MOI of 0.5 and incubated in a 5% CO_2_ incubator at 37 °C for 48 h; cells were then collected and subjected to two cycles of freezing and thawing (at −70 °C and RT) to break open the cells and release virions into the culture fluid. The suspension was then centrifugated at 3000× *g* and 4 °C for 5 min. The ASFV virions were concentrated by medium-speed centrifugation at 4 °C and 7000× *g* for 16 h. The precipitate was resuspended in 200 µL of cold 10 mM filtered Tris-HCl solution (pH 8.8).

The genomic DNA (gDNA) was extracted using a modified phenol-chloroform precipitation method. A total of 500 µL of resuspension buffer (10 mM Tris-HCl (pH 7.5), 0.25 units/µL of DNase, 20 µg/mL of RNase, 200 mM NaCl, 20 mM CaCl_2_, and 120 mM MgCl_2_) was added to the concentrated virus, then the mixture was incubated at 37 °C for 2 h. Subsequently, EDTA was added to a final concentration of 12 mM, and the mixture was kept for 10 min at 75 °C. To lyse the ASFV capsule and release the genome, proteinase K and SDS were added to a final concentration of 200 µg/mL and 0.5%, respectively, with further incubation at 45 °C for an hour. This was followed by adding “phenol/chloroform/isoamyl alcohol (25:24:1)” at a ratio of 1:1 (*v*/*v*), centrifugation at 3000× *g* for 3 min (at room temperature), and the collection of the upper aqueous phase. For gDNA precipitation, sodium acetate 3 M (pH 5) and 100% ice-cold alcohol (−20 °C) were added to the resulting aqueous phase at a ratio of 1:20 and 1:2 (*v*/*v*), respectively. The mixture was kept at −70 °C for at least 2 h, followed by centrifugation at 16,000× *g* + 4 °C for 30 min. The supernatant was discarded, and the pellet was washed with 300 µL of 70% alcohol and pelleted at 14,000× *g* for 5 min. The supernatant was decarded, and the pellet was dried at 55 °C for 15 min and resuspended in 50 µL of a 10 mM Tris solution (pH 8.8).

The quality of the total DNA was evaluated with a spectrophotometer used to measure the concentration of double-stranded DNA (ng) and absorption indices of A230/A260 and A260/A280, which shall be ≥2.0 and ≥1.8, respectively.

### 2.3. Sequencing

A total of 200 ng of purified DNA was fragmented, using a Covaris ME220 Focused-ultrasonicator (Covaris, LLC, Woburn, MA, USA) into fragments between 100 and 700 bp, with the majority ranging between 250 and 300 bp. Fragments approximately 300 bp in size were selectively purified using magnetic beads (SPRI), and 25 ng of DNA was processed according to the protocol of the MGIEasy Universal DNALibrary Prep Set (MGI Tech Co., Ltd., Shenzhen, China). This entailed blunt-end DNA fragment polishing and ligating adapters containing 10 bp single-end indexes. The DNBSEQ-400 platform MGI Tech Co., Ltd., Shenzhen, China). The paired-end 150 sequencing protocol was used to generate a dataset consisting of 148 million paired reads per sample [38].

Due to incorrect assembly algorithms for tandem repeats in the ASFV genome, additional sequencing of intergenic region I73R/*I329L* was performed by Sanger in all isolates. DNA samples were enriched using PCR, followed by purification and concentration of fragments according to the protocol described earlier [25].

### 2.4. Mapping Reads and Analyzing Sequences

The whole genome consensus sequences were assembled by mapping reads to a reference ASFV strain Georgia 2007/1 (NC_044959.2) in CLC Genomics v.9 Workbench package (https://www.pubcompare.ai/product/CyviCZIBPBHhf-iFV0P5/ (accessed on 30 November 2016)) using default settings. The open reading frames (ORF) were predicted using the Genome Annotation Transfer Utility (GATU) tool [39]. The obtained sequences were subjected to multiple alignments with those imported from GenBank (Appendix A) in the CLC Genomics v.9 Workbench package. The homology of the isolates was calculated using the NCBI nucleotide BLAST tool. Phylogenetic analysis was performed with Mega 11 software according to the recommended Tamura model with gamma distribution and considering invariant sites T92 + G + I (BIC = 522,851.1015; AICc = 521,158.4457) using Maximum Likelihood method with bootstrap 100 iterations [40].

## 3. Results

### 3.1. Evaluation of Sequencing and Assembling

Spectrophotometry (NanoDrop OneC, ThermoFisher Scientific, Waltham, MA, USA) of the gDNA showed that the DNA concentration for WGS was ≥200 ng/μL. Due to satisfactory quality (Figure 2), sequencing bioinformatic analysis was conducted. The whole genome sequence of all isolates studied in this paper has been confirmed and is published in Genbank (accession numbers are given in Appendix A).

Figure 2 shows that all the gDNA samples were satisfactorily purified from low molecular weight compounds, as the A230/A260 ratio corresponds to a reference value of ≥2.0. However, the A260/A280 ratio was 1.77–1.97 (reference range 1.8–2.0), which means an unsatisfactory purification of nucleic acid from protein impurities when preparing 7 out of 21 samples. The specific reads ranged from 1.7 to 25.47%, exceeding the recommended value of ≥1%. The mean sequencing coverage ranged from 132 to 1830 reads/nucleotide (>45).

The assembled sequences were mapped to the reference sequence Georgia 2007/1 strain. The length of the consensus sequences ranged between 187,596 and 190,594 bp, encoding 195 ORFs that were also predicted in all 21 studied isolates.

### 3.2. Analysis of Homology

To analyze the level of genome homology among all 21 isolates from central and eastern Russia, we used the most studied (i.e., reference) strains belonging to ASFV genotype II, i.e., the ones that had been recovered in various endemic countries of Eurasia (Georgia: the beginning of epidemic; Russia: the Kaliningrad region (“West”) and the Amur region (“East”); the Kabardino-Balkarian Republic (“Center”); Poland and Lithuania (POL/2015/Podlaskie and LT14/1490: the first outbreaks in eastern Europe); Moldova and Poland (Moldova 2017/1 and Pol17_55892_C754: the epidemic peak in Europe; China). The percentage of identity (%) was calculated and demonstrated as in the diagram in Figure 3.

As the diagram shows, the ASFV circulating in the central and eastern regions of Russia has a high homology level (99.95–99.99%) with genotype II strains registered in Eurasian countries. The lowest homology level is typical for four/five isolates from the Amur region (except for Amur_2022/WB-909–99.98–99.99%).

### 3.3. Phylogenetic Analysis

Based on the genome sequence of 56 ASFV strains and isolates (21 from this study and 35 imported from GenBank), phylogenetic relatedness was established (Figure 4).

As demonstrated in Figure 4, all isolates share the same origin, i.e., derived from a common ancestor (probably Georgia 2007/1 strain). Clade “Georgia 2007” (highlighted in blue) includes isolates that are closest to Georgia 2007/1 and includes isolates registered up to 2019 (inclusive) from Russia, Lithuania, and Poland.

All the isolates from central Russia (2019; 2021), first presented in this study, form the second clade, “Center of Russia” (Green). Within this clade, two subgroups, “Center of Russia 2021” and “Bryansk 2021”, can be additionally distinguished due to a divergence with a significant confidence level (bootstrap = 99).

Most (12/13) isolates recovered in the Far East of Russia share a common clade, “Asia” (yellow). This clade can be further divided into three groups: (i) “Amur 2022” (Amur_2022/WB-909 and Khabarovsk_2022/DP-1658); (ii) “Europe” (Moldova 2017/1, Belgium 2018/1 and Czech Republic 2017/1); and (iii) “Asia”, which includes the prevailing number of isolates from China and eastern Russia.

At the same time, the Khabarovsk_2021/WB-3967 isolate recovered from a wild boar in the eastern part of Russia phylogenetically belongs to the clade “Eastern Europe” (indicated in red), related to ASFV genetic variants from the Kaliningrad region of Russia, Poland, and Germany.

### 3.4. Analysis of SNPs

Multiple alignments revealed synonymous and non-synonymous SNPs, single- and multiple-oligonucleotide insertions, or deletions. Previously described SNPs, in addition to the newly identified ones, are shown in Table 1. At the same time, polyC, polyG regions, and highly variable genome fragments, mainly in terminal ends (5′ and 3′), are considered doubtful and thus not included in the results.

Thus, 8 out of 36 substitutions (22.22%) given in Table 1 turned out to be synonymous. A total of 25 out of 36 were non-synonymous (69.44%), leading to a change in the amino acid composition, and 3 out of 36 (8.33%) were located in intergenic regions and did not affect the primary sequence of the protein. In addition, six insertions were identified: a tandem repeat sequence (TRS), one single-nucleotide insertion in the intergenic region, and four single-nucleotide insertions in the ORFs, leading to a frameshift changing the protein’s primary structure. In addition, the deletion of nucleotide “A” was spotted in the *MGF 360-16R* gene, which induced another frameshift. The sequence of the *B646L* gene in all the studied isolates was identical to reference strain Georgia 2007/1. Thus, all isolates belong to genotype II. The genome of the isolates belonging to “Bryansk 2021” clade (Bryanskaya_2021/DP-18 and Bryanskaya_2021/DP-8823) contained seven unique substitutions (five transitions: “C” -> “T” in *MGF 110-3L* and F334L *genes*; “G” -> “A” in—*MGF 505-10R* and *MGF 360-15R* (two SNPs); two transversions “C” -> “G” in—*Q706L* and “G” -> “C” in—*NP1450L*).

For isolate Khabarovsk_2021/WB-3967, all genome polymorphisms, including those in previously characterized markers *MGF 110-7L*, *MGF 505-5R*, and *K145R* (Specific to Kaliningrad region) correlated with sequences from the Kaliningrad region of Russia. 

Eurasian strains from the clade “Georgia 2007” belong to the original genetic variant I (based on loci *MGF 110-1L*, *MGF 505-9R*, *NP419L*, and *I267L*), which has not been registered since 2019. Genetic variant II is typical for the isolates of other clades. Strain ASFV/Odintsovo 2014/WB is an exception (cluster “Georgia 2007”), belonging *to I267L-I*, *NP419L-II*, *MGF 110-1L-II*, and *MGF 505-9R-II*, as shown in Figure 5.

The genome of three isolates from the subgroup “Center of Russia 2021” (i.e., Belgorodskaya_2021/DP-11838; Permskyi_2021/DP-9916; Sverdlovskaya_2021/DP-9914) was found to have five clade-specific transitions (“C” -> “T” in genes *F1055L* and *QP383R*; “T” -> “C” in—*B962L*; “G” -> “A” in the ORF in *B169L* and *B354L*).

The non-synonymous transition “T” -> “C” in *MGF 360-10L* gene and “G” insertion in intergenic region *B602L/B385R* are unique to sort isolates in cluster “Asia” (Figure 6).

Some isolates from the clade “Asia” had a set of unique mutations:(1)Amur_2021/WB-10591, Amur_2021/WB-10595, Amur_2022/WB-905, Khabarovsk_2020/WB-11558 and Khabarovsk_2022/WB-1650 had two simultaneous substitutions “G” -> “A” in F778R gene and “C” -> “T” in intergenic region H233R/*H240R;*(2)Amur_2021/WB-10591, Amur_2021/WB-10595, Amur_2022/WB-905, and Khabarovsk_2022/WB-1650 are characterized by a common transition “T” -> “C” in C717R gene;(3)Amur_2021/WB-10591, Amur_2021/WB-10595, Amur_2022/WB-905 have 3 transitions “G” -> “A” in ORF *MGF 360-15R*, a “C” -> “T” in—*EP424R* and *B602L/B385R* and an “A” insertion in *MGF 505-2R*;(4)Amur_2021/WB-10595 and Amur_2022/WB-905 additionally have a substitution “A” -> “G” in *MGF 360-1La* locus and “A” insertion in A859L gene;(5)Isolates Amur_2022/WB-911, Khabarovsk_2020/DP-11562, Primorsky_2021/DP-9778 and Primorsky_2021/WB-9786 are characterized by insertion “C” in the ORF of *MGF 360-14L* and two transitions “C” -> “T” in the untranslated region of *ASFV G ACD 01990/DP60R* and CVR of *B602L gene* (as confirmed by the marker analysis during the previous research [28]). The division of Asia cluster into the above groups also clearly demonstrates the structure of the phylogenetic tree branches (Figure 4).

After whole genome consensus sequences were assembled and aligned, two 10-nucleotide TRSs (GAATATATAG) were detected in the IGR *I73R/I329L* in 21 isolates. However, the Sanger sequencing of this fragment shows that all the studied isolates contain three same-name TRSs (Figure 7).

The non-synonymous substitutions in the *E199L* gene described earlier have been identified in several isolates [41]: Transition “C” -> “T” in Amur_2021/WB-10591 and Odintsovo 2014/WB (position 167061); “C” -> “T”—JAO_2020/DP-6768, Leningrad_2019/WB-789, Kaliningrad 18/WB-12524, POL/2015/Podlaskie and LT14/1490 (position 167062); transversion “C” -> “G”—Khabarovsk_2022/DP-1658, Zabaykali 2020/WB-5314 and Zabaykaly_2020/DP-4905 (position 167188).

### 3.5. Comprehensive Analysis of Genome Markers

Considering characterized SNPs and phylogeny of 56 strains and isolates of ASFV genotype II, the sub-genotyping scheme was supplemented with widely used (CVR, IGR *I73R/I329L*, *MGF 505-9R/10R*, *K145R*, *O174L*, *MGF 505-5R*, *I267L*, *MGF 360-10L*) and alternative ones (*Q706L*, *B962L*), i.e., marker regions of the genome proposed in this study [10,25]. *Q706L-II* makes it possible to differentiate isolates from the Bryansk region, and *B962L-II*, an ASFV related to the clade “Center of Russia 2021”. The distribution of genetic variants and their classification into clades are given in Table 2.

Based on Table 2, a strictly differentiating set of marker areas is established for clusters “Georgia 2007”, “Eastern Europe”, “Asia”, “Bryansk 2021”, and “Center of Russia 2021”. However, the groups “Europe”, “Amur 2022”, and “Center of Russia” show a synonymous distribution of genetic variants.

## 4. Discussion

The ASF’s transboundary nature annually brings new disease-free countries from the Eurasian and American continents into the epidemic process [42,43,44]. Phylogenetic analysis of isolates has been recently acknowledged as an important part of the epidemiological studies of ASFV circulation [8,45,46].

DNA sequencing serves as an indispensable tool for phylogenetics, allowing the output of “raw” reads [47]. The inverted terminal repeats and homopolymers in the ASFV genome make it difficult to assemble correct consensus sequences [48]. This study demonstrates the impressive prospects of MGI Tech for whole genome sequencing of DNA viruses with a long genome (about 190,000 bp), where the ASF pathogen is used as an example. Despite the insufficient purification of gDNA samples from proteins (A260/A280 in some cases was <1.8), the total number of reads in all 21 samples was high (from 8,634,742 to 21,352,232). The number of reads mapped to the reference genome (178,677 to 2,358,546), with a minimum mean coverage of 132 reads/nucleotide, is enough for virus genome assembly. Full genome coverage (100%) contributed to the high-quality assembly of consensus genome sequences and, as a result, ensured accurate detection of SNP. Unfortunately, the assembly incorrectly determines the number of TRSs in IGR I73R/*I329L* (Figure 7); therefore, this fragment shall be subjected to Sanger sequencing.

Twenty-one isolates were studied during the analysis (eight from the central (2019; 2021) and thirteen (2020–2022) from the eastern regions of Russia). The 99.95–99.99% homology level with the known genotype II ASFV strains confirms the moderate mutation rate of the pathogen, which, according to accounts for 1.14 × 10^−5^ substitutions/site/year [46]. The high homology of the isolates may be related to their origin from one common ancestor (strain Georgia 2007/1). Interestingly, ASFV genotype II, recently isolated in Africa (Tanzania, Nigeria, and Ghana), shows the same degree of identity (99.95–99.96%) with strains from Georgia, Poland, and China [49,50,51]. Since a long sylvatic cycle of ASF transmission is typical for Tanzania, the role of *Ornithodoros* ticks in accelerating mutation rate and virus genetic variability may be overestimated [52]. Nevertheless, low genome variability and the need for epidemiological surveillance indicate the search for unique SNPs that facilitate the differentiation of isolates within genotype II [8].

In the Phylogenetic analysis of the whole genome sequence of 56 ASFV strains and isolates recovered in Eurasian countries, at least four large clades (clusters) have been formed, separated geographically and from the perspective of relatedness (except for “yellow”) (Figure 4).

The parent (initial, “blue”) cluster “Georgia 2007” has not been registered in Russia and Europe since 2019. Genome markers *MGF 110-1L-I*, *MGF 505-9R-I*, *NP419L-I*, and *I267L-I*, can be used to identify this cluster (Figure 5). We also recommend using a fragment of the *I267L* gene since all representatives of this clade, including the ASFV/Odintsovo 2014/WB strain, are grouped into *I267L-I*.

The “green” cluster, called the “Center of Russia”, includes all eight isolates from the central regions of the Russian Federation (Leningrad, Belgorod, Pskov, Bryansk, Sverdlovsk, and Perm regions). This means this clade has been registered in the country since 2019. No clade-specific mutations typical for all isolates of the “Center of Russia” were found. However, within this clade, there was a divergence with a high confidence level (99) of two groups, “Bryansk 2021” and “Center of Russia 2021”.

The “red” clade includes isolates from Eastern Europe: Poland, Germany, and the Kaliningrad region of Russia. However, Khabarovsk_2021/WB-3967 was also included in this group. SNP analysis and marker distribution of *K145R-III*, *MGF 505-5R-II*, and *MGF 110-7L-II* confirm this isolate belongs to a unique ASFV genetic variant from the Kaliningrad region [53,54]. Since transboundary transmission of ASFV from Kaliningrad region to Khabarovsk region through the wild boar is impossible (the distance is more than 7000 km), the probable reason for this virus variant presence in the East is the gradual involvement of various anthropogenic mechanisms. It should be noted that this variant was detected in a wild boar carcass in the region only in 2021 and has not spread further. At the same time, the situation with the spread of ASFV variants among wild boars in the areas of China bordering the Khabarovsk region and Asia, in general, is poorly studied and requires transparency from the supervisory authorities of the ASF-affected countries. The occurrence of recombinant ASFV variants in 2021–2023 in China, Russia, and Vietnam indicates the presence of active genetic interactions within the virus population detected in Asia.

Isolates from Moldova, the Czech Republic, and Belgium are phylogenetically close to ASFV circulating in Asia (particularly in China and eastern Russia) and are grouped into a “yellow” clade. Smaller subclusters such as “Amur 2022”, “Europe” (highlighted in orange), and “Asia” can be distinguished within it. The high homology level between the “European” and “Asian” isolates is explained by the probable introduction of ASFV genotype II to China (2018) from Western Europe [8,45]. The clade “Amur 2022” consists of 2 sequences from this study, i.e., Amur_2022/WB-909 and Khabarovsk_2022/DP-1658, which do not have differentiation SNPs, as well as representatives of the group “Europe”. Further search for genome variable regions shall be continued. Nevertheless, the heterogeneity of Asian isolates is significant, for which genome markers like *MGF 360-10L-II* and *B602L/B385R-II* can act as universal markers for genetic differentiation (Figure 6). In addition, three lines are identified within the Asia group: the first one includes five isolates (Amur_2021/WB-10591, Amur_2021/WB-10595, Amur_2022/WB-905, Khabarovsk_2020/WB-11558, and Khabarovsk_2022/WB-1650); the second one includes four (Amur_2022/WB-911, Khabarovsk_2020/DP-11562, Primorsky_2021/DP-9778 and Primorsky_2021/WB-9786); and the third one includes Zabaykali 2020/WB-5314 and Zabaykaly_2020/DP-4905. *F778R-II* and *H233R/H240R-II* regions are used to differentiate the first line of Asia, as they have a higher resolution; the second line is differentiated with the previously characterized CVR-*XIII* variant [27]. The third line can be referred to as “Mongolia 2019” since the isolates from the Zabaykalsky region described in the previous studies belong to the same group as SS-3/Mongolia/2019 [41,55,56].

Spread of the ASFV from various genetic taxa in the west and east of Eurasia is shown in Figure 8.

The study of the *E199L* gene polymorphism *is* of no interest to molecular epidemiology and to studying “clonal development” patterns of the epidemic process (Table 1). The high frequency of single-position mutations in this ORF in geographically uncorrelated isolates is associated with the virus adaptation to replication in similar idiotypic compartments since the protein encoded by this gene is responsible for activating the autophagy of the infected cells [57].

This work has shed light on the pattern of ASFV circulation in East Asia, particularly in eastern regions of Russia. Due to the disease spread in China, Vietnam, and South Korea, the results of phylogenetic analysis may be useful for other research teams conducting epidemiological studies [58].

## Figures and Tables

**Figure 1 viruses-16-01907-f001:**
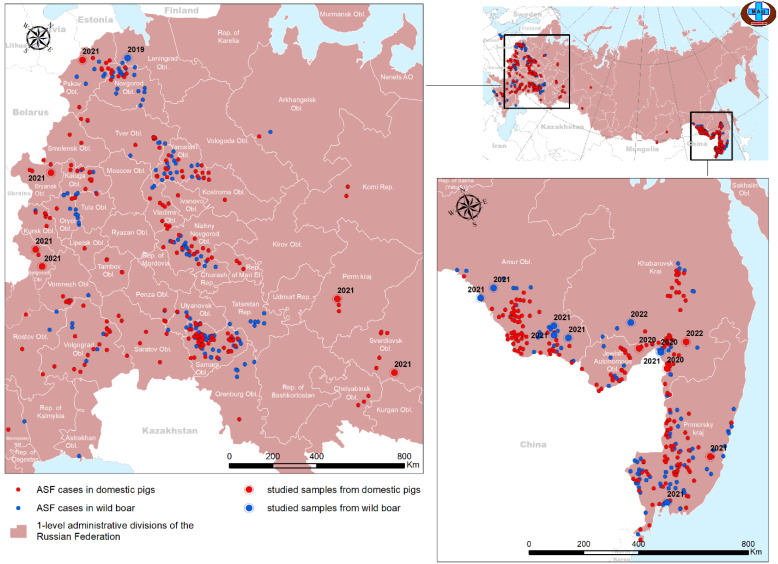
Epidemic situation on ASF in the Russian Federation from 2019 to 2022. The isolates covered by this study are indicated in large print.

**Figure 2 viruses-16-01907-f002:**
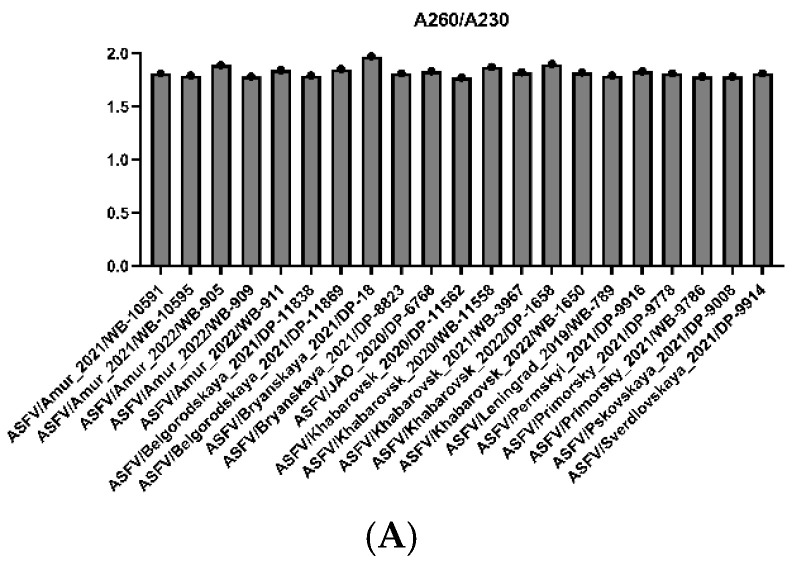
The ratio of absorbance at A260/A230 (low molecular weight purification rate, reference ≥ 2.0) (**A**), number of reads (**B**), A260/A280 (protein purification rate, reference ≥ 1.8) wavelengths for gDNA samples (**C**), number of ASFV specific reads (**D**), and mean coverage (**E**).

**Figure 3 viruses-16-01907-f003:**
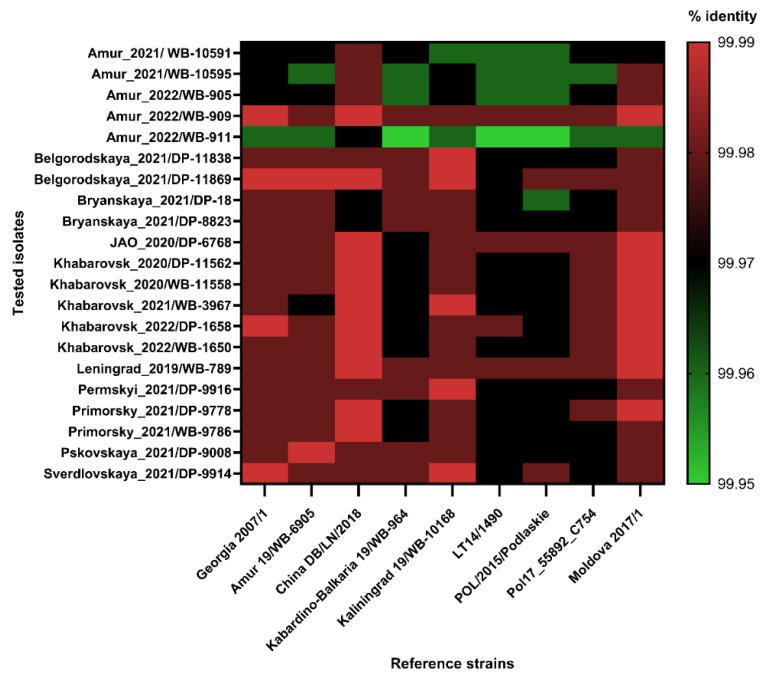
Percent of identity between the studied isolates and ASFV reference strains characterized in geographically separated areas of Eurasia.

**Figure 4 viruses-16-01907-f004:**
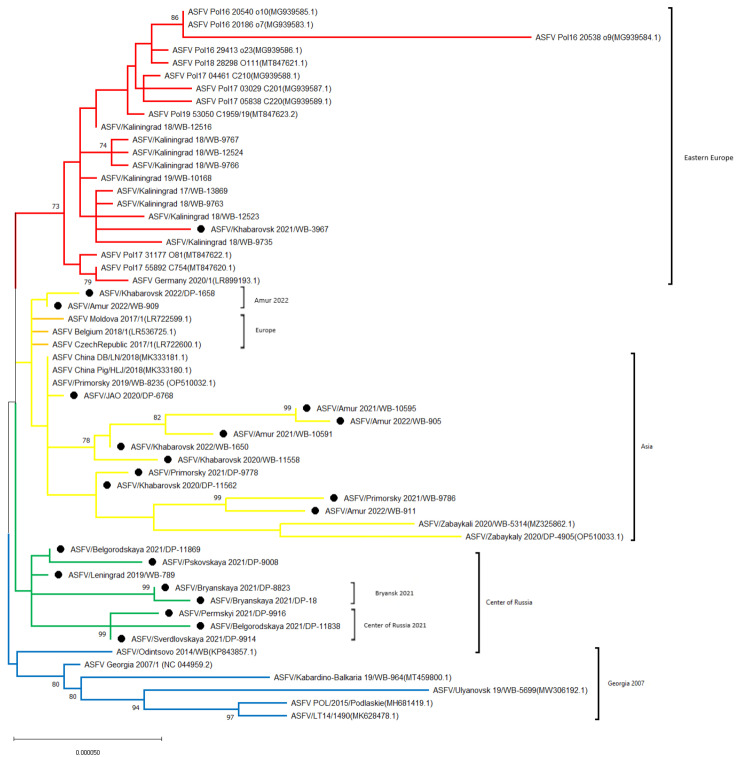
Phylogram showing relatedness between the twenty-one isolates from this study and the most well-known ASFV reference strains from Genbank. Note: the isolates covered by this research are labeled with •.

**Figure 5 viruses-16-01907-f005:**
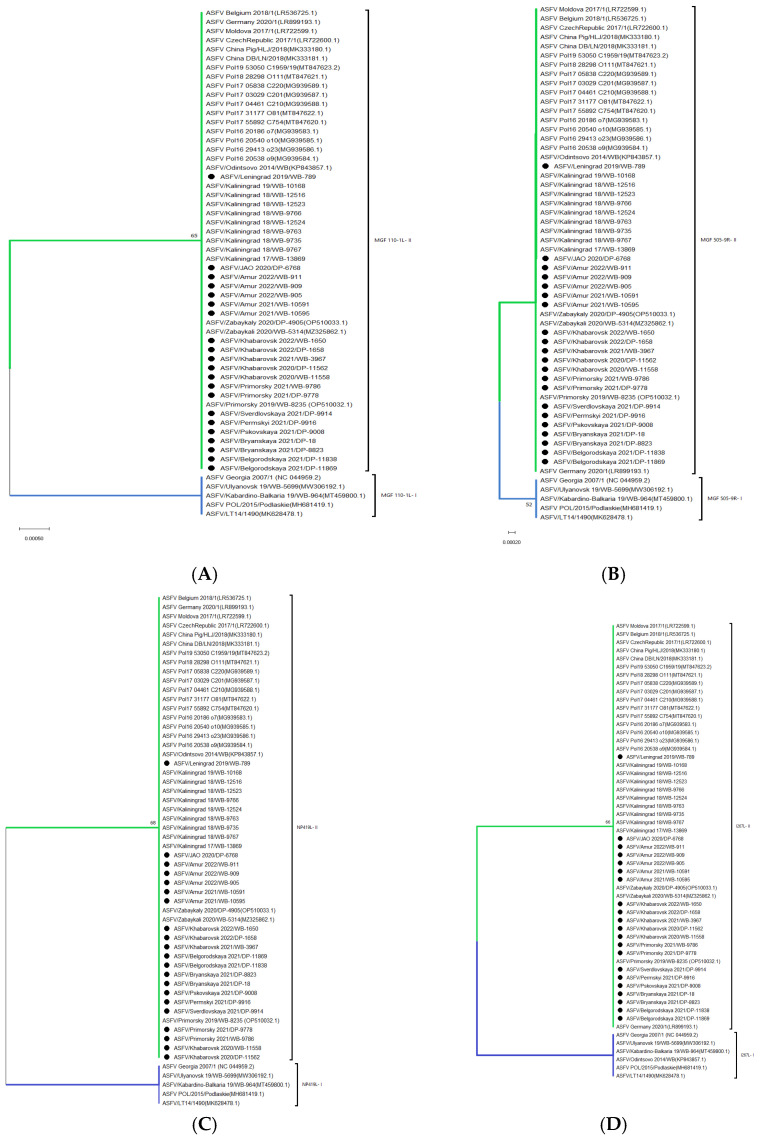
Maximum likelihood phylogenetic tree indicating the relationship of ASFV genotype II sequences, using selected ORFs MGF 110-1L (**A**), MGF 505-9R (**B**), NP419L, (**C**) and I267L (**D**). Note: the isolates covered in this study are labeled with a black dot; the original (root) and Genetic variant (I) are marked with blue while Genetic variant (II) is green.

**Figure 6 viruses-16-01907-f006:**
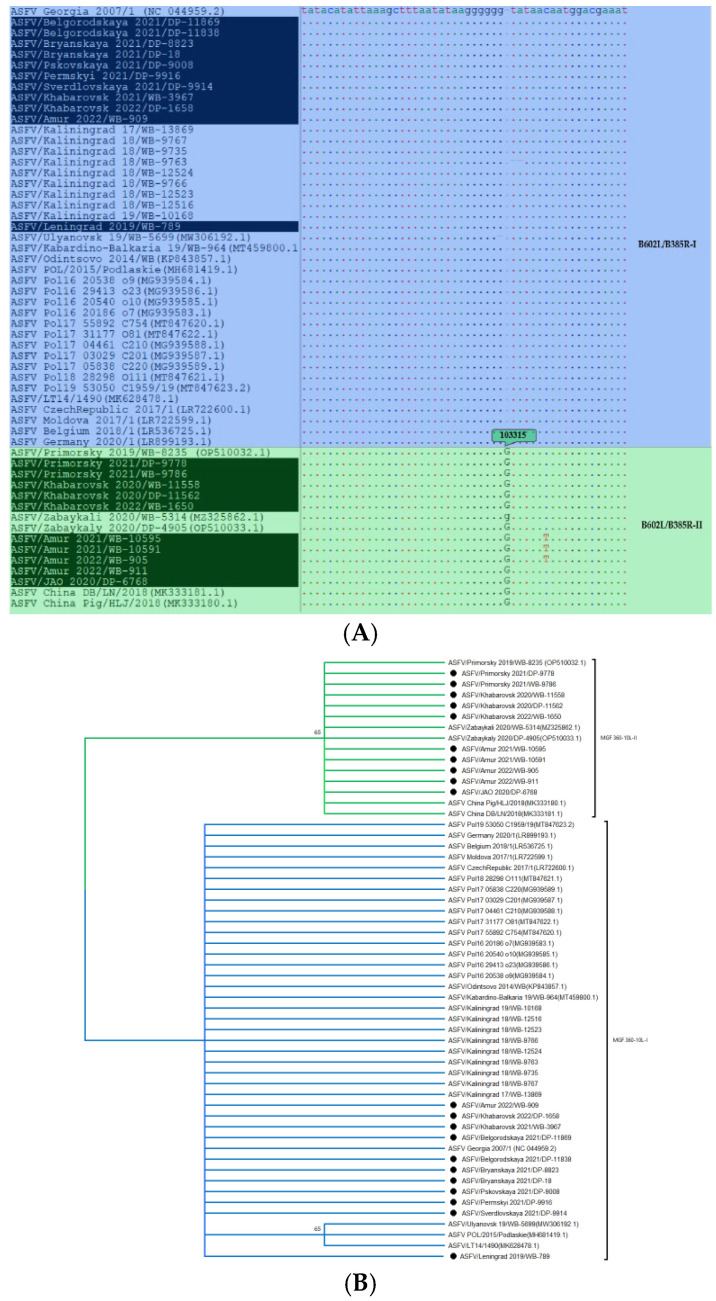
(**A**) Nucleotide sequence alignment of the intergenic region *B602L/B385R* of ASFV genotype II isolates, showing the formation of two variants. (**B**) Maximum likelihood phylogenetic tree of showing the relation between ASFV isolates based on the analysis of MGF *360-10L*. Variant II is indicated in green; other isolates belong to cluster Asia (blue). Isolates from this study are labeled with a black dot.

**Figure 7 viruses-16-01907-f007:**
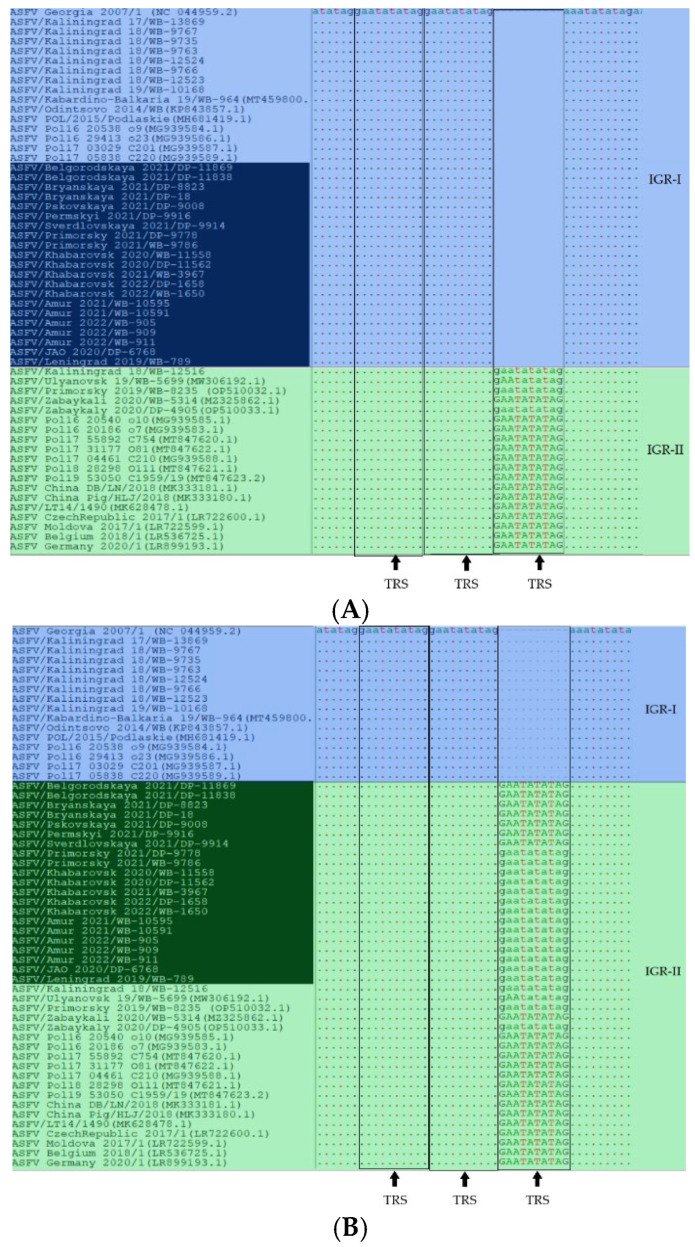
Alignment of IGR *I73R/I329L* sequences after genome assembly (**A**) and after Sanger sequencing (**B**). Note: The blue color indicates the IGR-I variant; the green color indicates IGR-II; the isolates covered in this study are labeled black background color.

**Figure 8 viruses-16-01907-f008:**
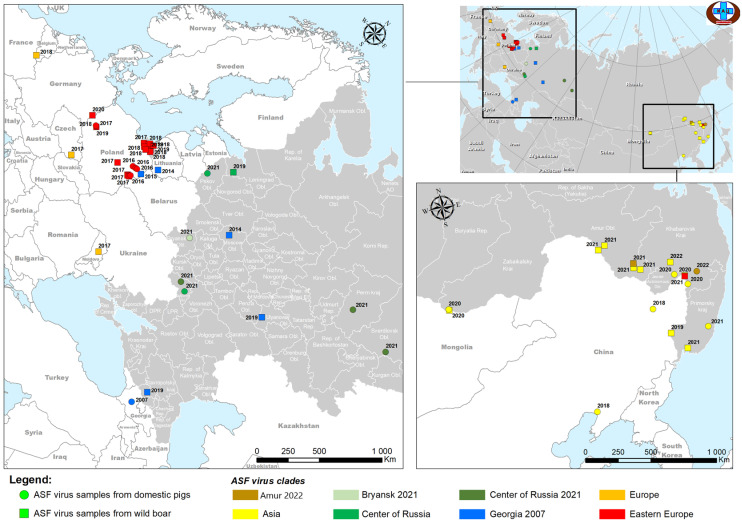
The molecular and epidemiological pattern of ASFV genotype II circulation in Europe and Asia, based on whole genome analysis of isolates.

**Table 1 viruses-16-01907-t001:** Comparative analysis of ORFs and amino acid changes in ASFV isolates of Genotype II.

Position (Strain Georgia 2007/1 NC_044959.2)	2106	7059	8342	10668	26425	33042/ 33043 32	35086/35087	36136	39306	44576	46557	50616	50655	50667	52924/52925	57933	58721	61821	66009	66152	72699	83138	95794	96624	101182	102721	103315/103316	103321	131463	134514	155937	156223	158805	161600	167061	167062	167188	170862	173338/173409	178010	184569	189902	190116/190117
ORF/Intergenic Region	MGF 360-1La	MGF 110-1L	MGF 110-3L	MGF 110-7L	MGF 360-10L	MGF 360-14L	MGF 505-2R	MGF 505-3R	MGF 505-5R	MGF 505-9R	MGF 505-10R	MGF 360-15R	MGF 360-15R	MGF 360-15R	A859L	F334L	F778R	F1055L	K145R	K145R	EP424R	C717R	B962L	B169L	B354L	B602L	IGR between B602L and B385R	IGR between B602L and B385R	NP1450L	NP419L	H233R	IGR between H233R and H240R	Q706L	QP383R	E199L	E199L	E199L	I267L	IGR I73R/I329L	MGF 360-16R	MGF 360-18R	IGR between ASFV G ACD 01990 and DP60R	DP60R
Amino Acid Variability	I = I	W -> stop	G -> D	S = S	N -> S	Frameshift	Frameshift	H -> R	V -> I	K -> G	R -> Q	D -> N	A -> T	E -> K	Frameshift	G -> K	Q = Q	Q = Q	Y = Y	S -> Y	F = F	S -> P	G = G	N = N	L -> F	S -> N	-	-	Q -> E	R -> S	V -> A	-	E -> Q	A -> V	G -> E	G -> R	A -> P	I -> F	-	Frameshift	A -> T	-	Frameshift
Georgia 2007/1	A	C	C	G	T	-	-	A	G	A	G	G	G	G	-	C	G	C	C	C	C	T	T	G	G	C	-	C	G	T	T	C	C	C	C	C	C	T	2TRS	G	G	C	-
ASFV/Amur_2021/WB-10591	A	T	C	G	C	-	A	G	G	G	G	A	G	G	-	C	A	C	C	C	T	C	T	G	G	C	G	T	G	C	T	T	C	C	T	C	C	C	3TRS	G	G	C	A
ASFV/Amur_2021/WB-10595	G	T	C	G	C	-	A	G	G	G	G	A	G	G	A	C	A	C	C	C	T	C	T	G	G	C	G	T	G	C	T	T	C	C	C	C	C	C	3TRS	G	G	C	A
ASFV/Amur_2022/WB-905	G	T	C	G	C	-	A	G	G	G	G	A	G	G	A	C	A	C	C	C	T	C	T	G	G	C	G	T	G	C	T	T	C	C	C	C	C	C	3TRS	G	G	C	A
ASFV/Amur_2022/WB-909	A	T	C	G	T	-	-	A	G	G	G	G	G	G	-	C	G	C	C	C	C	T	T	G	G	C	-	C	G	C	T	C	C	C	C	C	C	C	3TRS	G	A	C	A
ASFV/Amur_2022/WB-911	A	T	C	G	C	C	-	A	G	G	G	G	G	G	-	C	G	C	C	C	C	T	T	G	G	T	G	C	G	C	C	C	C	C	C	C	C	C	3TRS	-	G	T	A
ASFV/Belgorodskaya_2021/DP-11838	A	T	C	G	T	-	-	A	G	G	G	G	G	G	-	C	G	T	C	C	C	T	C	A	A	C	-	C	G	C	T	C	C	T	C	C	C	C	3TRS	G	G	C	A
ASFV/Belgorodskaya_2021/DP-11869	A	T	C	G	T	-	-	A	G	G	G	G	G	G	-	C	G	C	C	C	C	T	T	G	G	C	-	C	G	C	T	C	C	C	C	C	C	C	3TRS	G	A	C	A
ASFV/Bryanskaya_2021/DP-18	A	T	T	G	T	-	-	A	G	G	A	G	A	A	-	T	G	C	C	C	C	T	T	G	G	C	-	C	C	C	T	C	G	C	C	C	C	C	3TRS	G	G	C	A
ASFV/Bryanskaya_2021/DP-8823	A	T	T	G	T	-	-	A	G	G	A	G	A	A	-	T	G	C	C	C	C	T	T	G	G	C	-	C	C	C	T	C	G	C	C	C	C	C	3TRS	G	G	C	A
ASFV/JAO_2020/DP-6768	A	T	C	G	C	-	-	A	G	G	G	G	G	G	-	C	G	C	C	C	C	T	T	G	G	C	G	C	G	C	T	C	C	C	C	T	C	C	3TRS	G	G	C	A
ASFV/Khabarovsk_2020/DP-11562	A	T	C	G	C	C	-	A	G	G	G	G	G	G	-	C	G	C	C	C	C	T	T	G	G	T	G	C	G	C	T	C	C	C	C	C	C	C	3TRS	-	G	T	A
ASFV/Khabarovsk_2020/WB-11558	A	T	C	G	C	-	-	G	G	G	G	G	G	G	-	C	A	C	C	C	C	T	T	G	G	C	G	C	G	C	T	T	C	C	C	C	C	C	3TRS	G	G	C	A
ASFV/Khabarovsk_2021/WB-3967	A	T	C	A	T	-	-	A	A	G	G	G	G	G	-	C	G	C	T	A	C	T	T	G	G	C	-	C	G	C	T	C	C	C	C	C	C	C	3TRS	G	G	C	A
ASFV/Khabarovsk_2022/DP-1658	A	T	C	G	T	-	-	A	G	G	G	G	G	G	-	C	G	C	C	C	C	T	T	G	G	C	-	C	G	C	T	C	C	C	C	C	G	C	3TRS	G	A	C	A
ASFV/Khabarovsk_2022/WB-1650	A	T	C	G	C	-	-	G	G	G	G	G	G	G	-	C	A	C	C	C	C	C	T	G	G	C	G	C	G	C	T	T	C	C	C	C	C	C	3TRS	G	G	C	A
ASFV/Leningrad_2019/WB-789	A	T	C	G	T	-	-	A	G	G	G	G	G	G	-	C	G	C	C	C	C	T	T	G	G	C	-	C	G	C	T	C	C	C	C	T	C	C	3TRS	G	G	C	A
ASFV/Permskyi_2021/DP-9916	A	T	C	G	T	-	-	A	G	G	G	G	G	G	-	C	G	T	C	C	C	T	C	A	A	C	-	C	G	C	T	C	C	T	C	C	C	C	3TRS	G	G	C	A
ASFV/Primorsky_2021/DP-9778	A	T	C	G	C	C	-	A	G	G	G	G	G	G	-	C	G	C	C	C	C	T	T	G	G	T	G	C	G	C	T	C	C	C	C	C	C	C	3TRS	-	G	T	A
ASFV/Primorsky_2021/WB-9786	A	T	C	G	C	C	-	A	G	G	G	G	G	G	-	C	G	C	C	C	C	T	T	G	G	T	G	C	G	C	C	C	C	C	C	C	C	C	3TRS	-	G	T	A
ASFV/Pskovskaya_2021/DP-9008	A	T	C	G	T	-	-	A	G	G	G	G	G	G	-	C	G	C	C	C	C	T	T	G	G	C	-	C	G	C	T	C	C	C	C	C	C	C	3TRS	G	A	C	A
ASFV/Sverdlovskaya_2021/DP-9914	A	T	C	G	T	-	-	A	G	G	G	G	G	G	-	C	G	T	C	C	C	T	C	A	A	C	-	C	G	C	T	C	C	T	C	C	C	C	3TRS	G	G	C	A
ASFV/Primorsky_2019/WB-8235	A	T	C	G	C	-	-	A	G	G	G	G	G	G	-	C	G	C	C	C	C	T	T	G	G	C	G	C	G	C	T	C	C	C	C	C	C	C	3TRS	G	G	C	A
ASFV/Zabaykali 2020/WB-5314	A	T	C	G	C	-	-	A	G	G	G	G	G	G	-	C	G	C	C	C	C	T	T	G	G	C	G	C	G	C	T	C	C	C	C	C	G	C	3TRS	G	G	C	A
ASFV/Zabaykaly_2020/DP-4905	A	T	C	G	C	-	-	A	G	G	G	G	G	G	-	C	G	C	C	C	C	T	T	G	G	C	G	C	G	C	T	C	C	C	C	C	G	C	3TRS	G	G	C	A
ASFV/Kaliningrad 18/WB-12524	A	T	C	A	T	-	-	A	A	G	G	G	G	G	-	C	G	C	T	A	C	T	T	G	G	C	-	C	G	C	T	C	C	C	C	T	C	C	2TRS	G	G	C	A
ASFV/Kaliningrad 18/WB-9735	A	T	C	A	T	-	-	A	A	G	G	G	G	G	-	C	G	C	T	A	C	T	T	G	G	C	-	C	G	C	T	C	C	C	C	C	C	C	2TRS	G	G	C	A
ASFV/Ulyanovsk 19/WB-5699	A	C	C	G	T	-	-	A	G	A	G	G	G	G	-	C	G	C	C	C	C	T	T	G	G	C	-	C	G	T	T	C	C	C	C	C	C	T	3TRS	G	G	C	-
ASFV/Kabardino-Balkaria 19/WB-964	A	C	C	G	T	-	-	A	G	A	G	G	G	G	-	C	G	C	C	C	C	T	T	G	G	C	-	C	G	T	T	C	C	C	C	C	C	T	2TRS	G	G	C	-
ASFV/Odintsovo 2014/WB	A	T	C	G	T	-	-	A	G	G	G	G	G	G	-	C	G	C	C	C	C	T	T	G	G	C	-	C	G	C	T	C	C	C	T	C	C	T	2TRS	G	G	C	-
ASFV POL/2015/Podlaskie	A	C	C	G	T	-	-	A	G	A	G	G	G	G	-	C	G	C	C	C	C	T	T	G	G	C	-	C	G	T	T	T	C	C	C	T	C	T	2TRS	G	G	C	-
ASFV Pol17_55892_C754	A	T	C	A	T	-	-	A	A	G	G	G	G	G	-	C	G	C	C	A	C	T	T	G	G	C	-	C	G	C	T	C	C	C	C	C	C	C	3TRS	G	G	C	A
ASFV China DB/LN/2018	A	T	C	G	C	-	-	A	G	G	G	G	G	G	-	C	G	C	C	C	C	T	T	G	G	C	G	C	G	C	T	C	C	C	C	C	C	C	3TRS	G	G	C	A
ASFV China Pig/HLJ/2018	A	T	C	G	C	-	-	A	G	G	G	G	G	G	-	C	G	C	C	C	C	T	T	G	G	C	G	C	G	C	T	C	C	C	C	C	C	C	3TRS	G	G	C	A
ASFV/LT14/1490	A	C	C	G	T	-	-	A	G	A	G	G	G	G	-	C	G	C	C	C	C	T	T	G	G	C	-	C	G	T	T	T	C	C	C	T	C	T	3TRS	G	G	C	-
ASFV Moldova 2017/1	A	T	C	G	T	-	-	A	G	G	G	G	G	G	-	C	G	C	C	C	C	T	T	G	G	C	-	C	G	C	T	C	C	C	C	C	C	C	3TRS	G	G	C	A
ASFV Belgium 2018/1	A	T	C	G	T	-	-	A	G	G	G	G	G	G	-	C	G	C	C	C	C	T	T	G	G	C	-	C	G	C	T	C	C	C	C	C	C	C	3TRS	G	G	C	A
ASFV Germany 2020/1	A	T	C	A	T	-	-	A	A	G	G	G	G	G	-	C	G	C	C	A	C	T	T	G	G	C	-	C	G	C	T	C	C	C	C	C	C	C	3TRS	G	G	C	-

Note: The different colors indicate the bases: green—A; yellow—G; red—T; blue—C.

**Table 2 viruses-16-01907-t002:** Correlation between genetic variants of ASFV genome marker and phylogenetic clades.

Marker Fragment	CVR (Ген B602L)	IGR I73R/I329L	MGF 505-9R/10R	K145R	O174L	MGF 505-5R	I267L	MGF 360-10L	Q706L	B962L	Clade
Georgia 2007/1	I	I	I	I	I	I	I	I	I	I	Georgia 2007
ASFV/Amur_2021/WB-10591	I	II	I	I	I	I	II	II	I	I	Asia
ASFV/Amur_2021/WB-10595	I	II	I	I	I	I	II	II	I	I	Asia
ASFV/Amur_2022/WB-905	I	II	I	I	I	I	II	II	I	I	Asia
ASFV/Amur_2022/WB-909	I	II	I	I	I	I	II	I	I	I	Amur 2022
ASFV/Amur_2022/WB-911	XIII	II	I	I	I	I	II	II	I	I	Asia
ASFV/Belgorodskaya_2021/DP-11838	I	II	I	I	I	I	II	I	I	II	Center of Russia 2021
ASFV/Belgorodskaya_2021/DP-11869	I	II	I	I	I	I	II	I	I	I	Center of Russia
ASFV/Bryanskaya_2021/DP-18	I	II	I	I	I	I	II	I	II	I	Bryansk 2021
ASFV/Bryanskaya_2021/DP-8823	I	II	I	I	I	I	II	I	II	I	Bryansk 2021
ASFV/JAO_2020/DP-6768	I	II	I	I	I	I	II	II	I	I	Asia
ASFV/Khabarovsk_2020/DP-11562	XIII	II	I	I	I	I	II	II	I	I	Asia
ASFV/Khabarovsk_2020/WB-11558	I	II	I	I	I	I	II	II	I	I	Asia
ASFV/Khabarovsk_2021/WB-3967	I	II	I	III	I	II	II	I	I	I	Eastern Europe
ASFV/Khabarovsk_2022/DP-1658	I	II	I	I	I	I	II	I	I	I	Amur 2022
ASFV/Khabarovsk_2022/WB-1650	I	II	I	I	I	I	II	II	I	I	Asia
ASFV/Leningrad_2019/WB-789	I	II	I	I	I	I	II	I	I	I	Center of Russia
ASFV/Permskyi_2021/DP-9916	I	II	I	I	I	I	II	I	I	II	Center of Russia 2021
ASFV/Primorsky_2021/DP-9778	XIII	II	I	I	I	I	II	II	I	I	Asia
ASFV/Primorsky_2021/WB-9786	XIII	II	I	I	I	I	II	II	I	I	Asia
ASFV/Pskovskaya_2021/DP-9008	I	II	I	I	I	I	II	I	I	I	Center of Russia
ASFV/Sverdlovskaya_2021/DP-9914	I	II	I	I	I	I	II	I	I	II	Center of Russia 2021
ASFV/Primorsky_2019/WB-8235	I	II	I	I	I	I	II	II	I	I	Asia
ASFV/Zabaykali 2020/WB-5314	I	II	I	I	I	I	II	II	I	I	Asia
ASFV/Zabaykaly_2020/DP-4905	I	II	I	I	I	I	II	II	I	I	Asia
ASFV/Kaliningrad 17/WB-13869	I	I	I	III	I	II	II	I	I	I	Eastern Europe
ASFV/Kaliningrad 18/WB-9734	I	I	I	III	I	II	II	I	I	I	Eastern Europe
ASFV/Kaliningrad 18/WB-9763	I	I	I	III	I	II	II	I	I	I	Eastern Europe
ASFV/Kaliningrad 18/WB-9766	I	I	I	III	I	II	II	I	I	I	Eastern Europe
ASFV/Kaliningrad 18/WB-12523	I	I	I	III	I	II	II	I	I	I	Eastern Europe
ASFV/Kaliningrad 18/WB-12516	I	II	I	III	I	II	II	I	I	I	Eastern Europe
ASFV/Kaliningrad 18/WB-12524	I	I	I	III	I	II	II	I	I	I	Eastern Europe
ASFV/Kaliningrad 18/WB-9735	I	I	I	III	I	II	II	I	I	I	Eastern Europe
ASFV/Kaliningrad 19/WB-10168	I	I	I	III	I	II	II	I	I	I	Eastern Europe
ASFV/Ulyanovsk 19/WB-5699	V	II	I	I	I	I	I	I	I	I	Georgia 2007
ASFV/Kabardino-Balkaria 19/WB-964	I	I	I	I	I	I	I	I	I	I	Georgia 2007
ASFV/Odintsovo 2014/WB	I	I	I	I	I	I	I	I	I	I	Georgia 2007
ASFV POL/2015/Podlaskie	I	I	I	I	I	I	I	I	I	I	Georgia 2007
ASFV Pol16_20538_o9	I	I	I	II	I	II	II	I	I	I	Eastern Europe
ASFV Pol16_29413_o23	I	I	I	II	I	II	II	I	I	I	Eastern Europe
ASFV Pol16_20540_o10	I	II	I	II	I	II	II	I	I	I	Eastern Europe
ASFV Pol16_20186_o7	I	II	I	II	I	II	II	I	I	I	Eastern Europe
ASFV Pol17_31177_O81	I	II	I	II	II	II	II	I	I	I	Eastern Europe
ASFV Pol17_04461_C210	I	II	I	II	I	II	II	I	I	I	Eastern Europe
ASFV Pol17_03029_C201	I	I	I	II	II	II	II	I	I	I	Eastern Europe
ASFV Pol17_05838_C220	I	I	I	II	I	II	II	I	I	I	Eastern Europe
ASFV Pol18_28298_O111	I	II	I	II	I	II	II	I	I	I	Eastern Europe
ASFV Pol19_53050_C1959/19	I	II	I	II	II	II	II	I	I	I	Eastern Europe
ASFV Pol17_55892_C754	I	II	I	II	II	II	II	I	I	I	Eastern Europe
ASFV China DB/LN/2018	I	II	I	I	I	I	II	II	I	I	Asia
ASFV China Pig/HLJ/2018	I	II	I	I	I	I	II	II	I	I	Asia
ASFV/LT14/1490	I	II	I	I	I	I	I	I	I	I	Georgia 2007
ASFV CzechRepublic 2017/1	I	II	I	I	I	I	II	I	I	I	Europe
ASFV Moldova 2017/1	I	II	I	I	I	I	II	I	I	I	Europe
ASFV Belgium 2018/1	I	II	I	I	I	I	II	I	I	I	Europe
ASFV Germany 2020/1	I	II	I	II	II	II	II	I	I	I	Eastern Europe

## Data Availability

The datasets presented in this study were submitted to the Genbank database (PP982225–PP982245). The names of the isolates and accession number(s) can be found in the Appendix A.

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
