# Peer review of "Unique Nucleotide Polymorphism of African Swine Fever Virus Circulating in East Asia and Central Russia"

_viruses, 2024, doi:10.3390/v16121907_

Round 1

Reviewer 1 Report

Comments and Suggestions for Authors

The review points are included in the attachment.

Author Response

We thank the reviewer for the high level of our manuscript review.

Comments 1: Division into thematic paragraphs: The text is quite long and contains a lot of information. Introducing clear divisions into paragraphs will make it easier to read and understand. For example: • One paragraph for a general introduction to the ASF problem. • A second paragraph discussing current research approaches and their limitations. • A third paragraph describing specific challenges in Eurasia and China. • A fourth paragraph outlining the objective of the study

Response 1: We agree. The «Introduction» text is divided into 5 paragraphs: 1- problem statement, its description and the importance of differentiation of isolates and understanding the spatial and temporal pattern of ASF spread; 2 paragraph - existing approaches to genotyping and sub-genotyping, their implementation, and shortcomings in them (using the example of recombinant detection); 3 - the problem of ASF virus sub-genotyping in Eurasia in general; 4 - the problem of the East (in particular, China, Mongolia and Russia); 5 - the final problem and goal statement.

Comments 2: Clear Formulation of the Study Objective: • The study's objective is mentioned at the end of the introduction, but it could be highlighted more effectively, e.g., in a separate sentence or paragraph. • It would be beneficial to add why detecting SNPs and conducting phylogenetic analysis are crucial for monitoring and controlling ASF

Response 2: We agree. We emphasized the importance of phylogenetic analysis in the 1st paragraph, where we talked about the impact of understanding the spatio-temporal pattern of ASF spread in epidemiological surveillance. The purpose of the study was put in the last sentence of «Introduction». The problem is summarized before the purpose statement.

Comments 3: Style and Language: • Avoiding generalizations: Some parts of the text (e.g., "which are rapidly spreading in Asian countries") could be replaced with precise statements, providing examples or mechanisms. • Standardizing terminology: Different terms are used in various parts of the introduction to refer to the same concept (e.g., "unique SPNs" and "SNPs"). It is better to use consistent terminology throughout

Response 3: We have tried to avoid of generalizations and come to a single term “SNPs”.

Comments 4: Global Context: • Add a brief mention of the global significance of ASF to highlight its impact beyond the studied region (e.g., Europe, Asia). • It is important to mention why ASF is a key issue on a global scale, such as its impact on international trade and the economic consequences of the disease.

Response 4: We don't quite agree that the article is global in terms of economics. The manuscript focuses on the search for substitutions and indels in the ASFV genome, which may help in international epidemiologic surveillance. However, the object of her research is limited to Eurasian countries.  Africa and the Caribbean countries are beyond the scope of this article. As for examples of economic losses, they are added in the 1st paragraph of «Introduction».

Comments 5: Accuracy of laboratory method descriptions: • The use of the term "homogenization" is correct, but it would be helpful to specify exactly how the homogenization was carried out (e.g., using what equipment, at what speed). • It would be important to indicate the type of 0.45-micron filter used (e.g., material, manufacturer) to avoid potential discrepancies in results due to differences in equipment.

Response 5: We have added information to the manuscript about a 0.45 micropore size bacterial filter (Corning, USA). As for homogenization, tissues and organs were ground in a mortar and then phosphate-buffered saline was added, bringing it to 10% suspension. We made a more correct description in the text: “A 10% suspension (w/v) was prepared from samples using a phosphate-buffered saline (PBS) solution, suspensions were then clarified by centrifugation at 1000 g for 3 minutes”.

Comments 6: More precise description of PBMP cell preparation: • The cell culture process description could be more detailed (e.g., culture conditions, medium, incubation time). It would also be useful to add the conditions under which PBMP cells were cultured prior to infection (e.g., temperature, CO2 levels, incubation time before infection).

Response 6: All studies related to cell culture preparation and ASFV infection are prescribed in our previously published protocol, which is referenced in the text [Puzankova O. et al., 2023]. This paper unified the method we have been using for all ASFV isolates.

Comments 7: Description of the identification and analysis process: • "Dense hemadsorption" could be described in more detail: what does this phenomenon mean in the context of ASFV? Were additional tests used to confirm the infection? • It would be helpful to state whether control viruses or negative samples were used to verify the accuracy of the procedure.

Response 7: Only the hemadsorption test was used to detect virus propagation. The term “dense” in the context of hemadsorption has been deleted: we agree it is inaccurate. As for controls, of course we use positive (strain Arm 07) and negative samples in routine ASF diagnosis. Their use is clearly stated in the FAO manual for ASF diagnosis and also suggested in the protocol published earlier [Puzankova O. et al., 2023]. In our opinion, in the context of this article, the method of virus isolation in PBMP cell culture is not original and interesting for the reader.

Comments 8: A more detailed description of the methodology is needed in paragraph number 2.3. Furthermore, I would suggest including the following points when revising the article: • Sequencing assembly software: Although it is mentioned that CLC Genomics v.9 Workbench was used, it would be helpful to specify which particular modules or settings were used for sequence assembly. For example, were any specific parameters or filters applied during the assembly process (e.g., read length, quality threshold)? • Versioning and reproducibility: Several tools and software versions (CLC Genomics v.9, Mega 11, GATU, NCBI BLAST) are mentioned, but it would be helpful to provide the exact version numbers to ensure the reproducibility of the experiment, especially if there are newer software versions that could affect the analysis results. This also applies to the reference ASFV strain Georgia 2007/1, as version numbers or accession numbers for databases (e.g., GenBank) were not provided.

Response 8: We agree. Added access and latest versions to programs where possible. Parameters for mapping reads to reference were used default (we prescribed this). Also the sequencing section was significantly changed and reformatted.

Comments 9: The sentence “ Spectrophotometry of the gDNA aliquots…” should be supplemented with the name and technology used for DNA measurement (paragraph 3.1.) 2. A230/A260 and A260/A280: Although it is mentioned that the A230/A260 ratio corresponds to the reference value of ≥ 2.0, and the A260/A280 ratio ranged from 1.77 to 1.97 (with the reference range of 1.8-2.0), it would be helpful to provide an explanation as to why the A260/A280 ratio is lower than optimal in some samples. It should also be clarified how this difference may impact further analysis (e.g., whether it affects sequencing quality and data analysis).

Response 9: We have added information about the spectrophotometer in paragraph 3.1. Values below the A280/A260 reference values are discussed in the 2nd paragraph of the Discussion. It says that despite the low values of purity from protein impurities the bioinformatic metadata satisfied us.

Comments 10: Contextualization of Results: • SNP Analysis: There’s a statement about "reduced mutation rate," but no further explanation about the biological significance or why this is noteworthy. Adding a sentence or two on what this means for virus evolution or transmission could help the reader understand the importance of this finding. • Homology: The statement that ASFV isolates from different regions have "99.95 99.99% homology" is important, but it could be useful to explain the implications of such a high identity. For example, does this suggest a lack of genetic drift over time, or is it expected for strains within the same clade?

Response 10: We have supplemented paragraph 3 of the Discussion with an explanation of what the high homology of ASFV strains may be related to. For the same reason, a moderate rate of mutations is manifested. The origin of isolates from a common ancestor and the ticks absence in Eurasia, which can increase mutation rate of ASFV, are given as assumptions.

Comment 11: Grammar and Structure: • Sentence Length: Some sentences are quite long and could be broken down into smaller, more digestible sentences. For example, "The "green" cluster, called "Center of Russia", includes all 8 isolates from the central regions of the Russian Federation (Leningrad, Belgorod, Pskov, Bryansk, Sverdlovsk and Perm regions) and they are all used in this study." This could be split into two: one describing the cluster and another mentioning the study. • Punctuation: There are instances where commas are missing, making sentences a bit harder to follow, such as in "Two isolates from group 'Bryansk 2021' have 7 specific substitutions (Table 1)." A comma after "2021" would help readability.

Response 11: We have tried to simplify and shorten the discussion text in the revised version of the manuscript.

Comment 12: Clade Identification: There is some inconsistency in the way clades are presented, such as using both "green" and "Center of Russia" interchangeably. It might be clearer to stick to one term throughout (e.g., consistently using "Center of Russia" or "green cluster"), or briefly explain the reason for the distinction between the terms. • Use of Regions and Years: When mentioning different regions and years, it would be helpful to consistently format the names and years for better readability. For example, use either "Amur_2022" or "Amur (2022)" but avoid mixing formats.

Response 12: We agree - the clade names are unified. The color of the clade is consistent with the color of the branch on the phylogenetic tree. In our opinion, it is visually easier to perceive the information than to refer to the name of the clade every time.

Comment 13: In some parts, you repeat certain ideas without adding new information. For instance, you mention "group 'Bryansk 2021'" and "group 'Center of Russia 2021'" multiple times. You can consolidate these references to avoid redundancy.

Response 13: We agree. We have tried to avoid repetition in the revised version of the manuscript.

Comments 14: At the end, there are references to various studies (e.g., "Shen Z. et al., 2022") without a clear explanation or citation format. It's important to provide a consistent and correct citation format for clarity, and if these studies are essential to the argument, you could briefly summarize their key contributions to your analysis.

Response 14: We have also tried to bring the citations to a common format.

Reviewer 2 Report

Comments and Suggestions for Authors

Authors reported single nucleotide polymorphisms in African swine fever virus genomes circulating in Central Russia and Eastern Asia. The study would be useful for the researchers in the field given the severity of ASFV disease in the Europe and Asia. The study is well designed and the manuscript is well written. This reviewer has following minor suggestions for the authors: 

Methods:

1. that shall be ≥ 2.0 and ≥ 1.8, respectively.

2. Please provide more information on library preparation, genome sequencing, and assembly. 

Results:

3. ≥ 200 ng. I believe the unit is per microliter. 

4. There are a few typing errors. Please read through the text carefully to correct them. 

Author Response

We thank the reviewer for his help in correcting inaccuracies in our manuscript. All your comments have been corrected. We have added information about library preparation to the Sequencing.

Kind regards,
team of authors

Please don't see the attachment. . File attached by mistake

Round 2

Reviewer 1 Report

Comments and Suggestions for Authors

I have no further comments. The article in its current form can be published.

Author Response

Thank you very much for reviewing the manuscript